# Outcomes of Endoscopic Endonasal Dacryocystorhinostomy in Glaucoma Patients

**DOI:** 10.3390/jpm14040348

**Published:** 2024-03-27

**Authors:** Gian Marco Pace, Francesco Giombi, Giovanna Muci, Gianmarco Giunta, Francesca Pirola, Egidio Serra, Jessica Zuppardo, Fabio Ferreli, Paolo Vinciguerra, Giuseppe Mercante, Alessandra Di Maria, Giuseppe Spriano, Luca Malvezzi

**Affiliations:** 1Department of Biomedical Sciences, Humanitas University, Via Rita Levi Montalcini 4, Pieve Emanuele, 20090 Milan, Italy; gianmarco.pace@humanitas.it (G.M.P.); giovannaquintina.muci@st.hunimed.eu (G.M.); gianmarco.giunta@st.hunimed.eu (G.G.); egidio.serra@humanitas.it (E.S.); jessica.zuppardo@humanitas.it (J.Z.); fabio.ferreli@hunimed.eu (F.F.); paolo.vinciguerra@hunimed.eu (P.V.); giuseppe.mercante@hunimed.eu (G.M.); alessandra.di_maria@humanitas.it (A.D.M.); giuseppe.spriano@hunimed.eu (G.S.); luca.malvezzi@humanitas.it (L.M.); 2Otorhinolaryngology Unit, IRCCS Humanitas Research Hospital, Via Manzoni 56, Rozzano, 20089 Milan, Italy; francescapirola.fp@gmail.com; 3Department of Ophthalmology, IRCCS Humanitas Research Hospital, Via Manzoni 56, Rozzano, 20089 Milan, Italy; 4Otorhinolaryngology Head and Neck Surgery Unit, Casa di Cura Humanitas San Pio X, Via Francesco Nava 31, 20159 Milan, Italy

**Keywords:** endoscopic dacryocystorhinostomy, epiphora, glaucoma, nasolacrimal duct obstruction, topical medications

## Abstract

Background: Anti-glaucoma eye drops have been investigated due to their production of fibrotic changes on the conjunctival surface, undermining the functioning of the upper lacrimal drainage system. We aimed to assess whether these effects may impair the effectiveness of endoscopic endonasal dacryocystorhinostomy (EE-DCR). Methods: This is a single-center observational retrospective study on EE-DCR via a posterior approach. Resolution of epiphora and dacryocystitis were analyzed after 1 (T1) and 6-months (T2) from surgery. Surgical success was defined as anatomical (patency at irrigation, no recurring dacryocystitis) or complete (zeroing of Munk score). Results: Twenty patients (32 sides) were enrolled. Preoperatively, 93.75% (*n* = 30/32) presented severe (Munk 3–4) epiphora and 68.75% (*n* = 22/32) recurrent dacryocystitis. At T1, 50.0% (*n* = 16/32) were referred with residual epiphora (Munk ≥ 1) and 18.75% (*n* = 6/32) dacryocystitis. At T2, 31.25% (*n* = 10/32) still complained of epiphora (Munk ≥ 1) and 6.25% (*n* = 2/32) dacryocystitis. Difference of outcomes at aggregate and paired timepoints (except for T1 versus T2) resulted in statistical significance (*p* < 0.05). At T2, 22 (68.75%) complete, 8 (25.0%) anatomical successes and 2 (6.25%) surgical failures were observed. Conclusions: Despite the chronic uptake of anti-glaucoma eye drops, EE-DCR guaranteed high rates of clinical relief from epiphora and remarkable decreases in the rates of recurrent dacryocystitis.

## 1. Introduction

Nasolacrimal duct obstruction (NLDO) is a widely spread clinical condition consisting of lacrimal drainage system dysfunction, resulting in a tear blockade, which eventually turns into epiphora, and episodes of lacrimal gland infections (e.g., dacryocystitis). Based on age at diagnosis, NLDO may be congenital or acquired, the latter being the most frequent worldwide [1]. Secondary nasolacrimal duct obstruction (SANDO) occurs whenever a specific event, such as lacrimal sac neoplasia, inflammatory diseases, infections, mechanical obstruction, or trauma, is responsible for the clinical scenario. This situation should be regularly ruled out before proceeding with the therapeutic process. In cases in which no specific etiopathological factors are advised, this condition is known as primary nasolacrimal duct obstruction (PANDO), which accounts for most cases [2]. According to recent evidence, PANDO is considered a multi-factorial disease, driven by both genetic and acquired predisposing conditions, such as vascular factors, local hormonal imbalance, microbial influence, nasal abnormalities, and autonomic dysregulation, which eventually turns into lacrimal drainage system narrowing and obstruction [3]. Dacryocystorhinostomy (DCR) is the mainstay of treatment for PANDO, allowing the surgeon to overcome tear drainage blockage by opening the lacrimal sac onto the lateral nasal wall through an external or endonasal approach. Dacryocystorhinostomy was first described as an open technique, which required a medial canthal facial skin incision that was hence called external-DCR (ex-DCR). Although ex-DCR has proven to reliably improve patients’ symptoms and related quality of life, disadvantages include medial canthal scar and disruption of the orbicularis, which may lead to an abnormal tear pump, as well as increased post-operation morbidity due to the skin incision [4]. After many years, in 1989 McDonogh and Miering presented the first endoscopic approach to the lacrimal sac [5]. Initially, the endonasal approach did not gain popularity until the more recent development of endoscopic endonasal surgery, which allowed the diffusion of endoscopic endonasal dacryocystorhinostomy (EE-DCR) and its establishment as a proven technique in the treatment of NLDO [6]. Up to date, several DCR surgical techniques have developed, but there is still lack of evidence on which could be the most effective.

Nasolacrimal duct obstruction may be further divided in upper NLDO (U-NLDO) if the obstacle is located at the level of punctum, canaliculus or common canaliculus, lower NLDO (L-NLDO). Most frequently, the site of obstruction is the junction between the lacrimal sac and the nasolacrimal duct [7]. Nevertheless, in many cases patients experience lacrimal drainage dysfunction in the presence of anatomical patency: to define this concept, in 1955 Demorest and Milder introduced the expression “functional block” [8]. The association between an NLDO and a U-NLDO, or with a functional block, represents a major cause of DCR failure and persisting epiphora, thus enhancing the relevance of adequate patient selection as a key point in obtaining a positive surgical outcome. In this regard, topical eye drops, in particular anti-glaucoma medications, have been investigated on producing fibrotic changes on the conjunctival surface, undermining the functioning of the upper lacrimal drainage system [9]. Recently, interest in the association between glaucoma and U-NLDO has been rising, but only a few articles with limited cohorts of patients have been published. Therefore, there is a growing focus on understanding whether patients with NLDO taking up glaucoma topical therapy can clinically benefit from DCR procedures. The inquiry revolves around whether the impaired functioning of the superior lacrimal system, significantly affected by anti-glaucoma medications, might impede symptom resolution, despite addressing lower tear obstruction. As a result, there is concern that patients may not achieve satisfactory improvement from such interventions. Mandal et al. first published a single-center experience reporting clinical post-operative outcomes in a cohort of patients under topical anti-hypertensive eye medications [10]. They observed that topical anti-glaucoma eye drops may lead to a higher failure rate for DCR surgery, possibly due to the provocation of an inflammatory cicatricial response. Nevertheless, in their research, the majority of patients (57.1%) were enrolled with DCR having been performed via an external approach [10]. Accordingly, this study aims to evaluate the outcomes of a cohort of glaucoma patients affected by NLDO treated with EE-DCR via a posterior lacrimal fossa approach in a tertiary referral Hospital Department, with a focus on results in terms of relief from epiphora and recurrent dacryocystitis.

## 2. Materials and Methods

### 2.1. Study Design

This is a single-center observational retrospective study. All consecutive EE-DCRs performed on patients affected by NLDO while topically treated for glaucoma between January 2020 and January 2023 in the Humanitas Clinical and Research Hospital were analyzed. The same ENT and ophthalmic surgeons performed all the procedures. Inclusion criteria also encompassed amenability to endoscopy and general anesthesia, age greater than 18 years, and informed consent. All enrolled patients were affected by primary open-angle glaucoma, defined by elevated intraocular pressure measured by tonometry in the presence of open anterior chamber angles, as stated by the American Academy of Ophthalmology [11]. Exclusion criteria were previous DCR and/or nasal surgery. The outcomes of the study were the assessment of post-operative persistent epiphora and recurrent dacryocystitis. This study was conducted in accordance with the ethical standards of the Declaration of Helsinki and its later amendments, and institutional ethics committee approval was obtained (IRCCS-ICH-IEC/2874).

### 2.2. Preoperative Evaluation

The diagnostic approach to NLDO included an ophthalmologic evaluation, with probing and irrigation of the lacrimal ways demonstrating the L-LDSO, an ENT evaluation with nasal endoscopy, and a preoperative dacryo-CT scan to confirm the site of obstruction.

### 2.3. Surgical Technique

The procedure is carried out under general anesthesia by an ENT surgeon and an ophthalmologist. With a 4 mm 0° rigid nasal endoscope, after nasal mucosa decongestion, retrograde inferior uncinectomy with backbite forceps is performed. Removal of the vertical portion of the uncinate process (UP) gives access to the lacrimal bone; whenever present, the cranial opening of a pneumatized agger nasi cell further allows a wider surgical view. From the perspective of an “anatomy-driven” surgical technique, we do not rely on a rigid light probe to locate the projection of the lacrimal sac on the lateral nasal wall. Therefore, after performing uncinectomy and agger nasi opening, the lacrimal bone (LB) is identified at the level of the middle turbinate axilla, confirming the relationship between the UP and the lacrimal bone previously evaluated by dacryo-CT scan. The LB is then skeletonized with a high-speed diamond burr. In this way, there is no need to prepare mucosal flaps, since the sac is approached posteriorly, at the thinnest point of the lateral nasal wall, and is exposed with little drilling, without removing parts of the frontal process of the maxilla and without leaving bony walls exposed (Figure 1).

An angled scalpel and a circular cutting punch are used to incise the sac and marsupialize its walls. Aims are the creation of as wide a stoma as possible and the correct exposure of the common internal punctum from the nasal cavity. Then, a bi-canalicular silicone stent is inserted through the stoma in order to guarantee post-surgical patency, and tied inside the nasal fossa. In our experience, we do not rely on the intraoperative adjuvant application of antimetabolites (e.g., mitomycin C) at the level of the nasolacrimal stoma during EE-DCR. Indeed, according to a recent analysis, mitomycin C does not translate into improved functional success in most of the published case series [12]. Finally, at the end of surgery, one small piece of cotton is placed in the middle meatus for hemostatic purposes and removed at patient’s discharge, normally 4 h after awakening from general anesthesia.

### 2.4. Post-Operative Care

To remove intranasal debris and to monitor surgical site healing, as well as the patency of the stoma, every patient was evaluated by an ENT and an ophthalmologist 7, 21 and 45 days after surgery. The bi-canalicular silicone stent was removed at least 3 weeks after surgery, when local healing had been obtained. For subjects with a greater tendency to scarring, the stent was maintained in place for a longer time. After surgery, the dosage of anti-glaucoma medications was kept unchanged compared to the pre-operative period. For statistical purpose, topical eye drops were coded based on their pharmacological class: β-blockers, prostaglandin analogues, carbonic anhydrase inhibitors and α_2_-agonists.

Post-operative outcomes were assessed by proper questionnaires. All patients were asked pre-operatively (T0) and at different follow-up times (1 month and 6 months post-op—T1 and T2, respectively) to rate the frequency of dacryocystitis as well as their tearing eyes, based on the 0–4 epiphora severity scale developed by Munk et al. [13], ranging from 0 (no watering) to 4 (constant watering). Dacryocystitis was defined by the clinical evidence of erythema and edema over the medial canthus and the area overlying the lacrimal sac at the inferomedial portion of the orbit, possibly associated with purulent discharge. The survey is available as supplemental content (Appendix A). Pre- and post-operative data were analyzed and compared. Surgical complete success was defined by the resolution of symptoms and patency on irrigation, a negative Munk Score and absence of dacryocystitis. Anatomical success was considered in case of patency on irrigation associated with persistency of mild to moderate epiphora, a Munk Score = 1–3 and absence of dacryocystitis. Surgical failure was defined as persistence of severe epiphora (Munk Score = 4), dacryocystitis recurrences and/or an inability to irrigate the lacrimal system post-operatively. Eventual revision surgery was indicated whenever a symptomatic scarring stenosis of the lacrimal stoma was still observed at T2 examination, despite medical therapy.

### 2.5. Data Analysis

Statistical analysis was conducted using IBM^®^ SPSS Software for Macintosh, Version 26.0 (IBM Corp., Armonk, NY, USA). SPSS (Version 28 for Macintosh, IBM^®^). The sample size was calculated with α-error set at 0.05 and β-error at 0.20, for a study power of 80%. Ordinal variables were expressed by numbers and percentages, whereas continuous non-parametrical data were presented by median and 25°−75° interquartile range (IQR). Differences between ordinal related samples were calculated with the Friedman test and results were presented for both aggregate and paired samples. The significance threshold was set as a *p*-value ≤ 0.05.

## 3. Results

Out of 172 EE-DCR performed in our institute during the period of observation, 20 patients (females: 80.0%, *n* = 16/20) were considered eligible, for a total of 32 operated sides (left eye: 56.25%, *n* = 18/32). The median age was 71.2 years (range: 59–86; IQR: 60.5–78.5). The minimum sample size to assess surgical outcomes at T1 and T2 was 14 and 11, respectively, and was thus satisfied by our study population. Over the studied period, all included subjects were under anti-glaucoma topical medication. By drug class, eye drops assumed by our cohort were the followings: β-blockers 80.0% (*n* = 16/20), prostaglandin analogues 60.0% (*n* = 12/20), carbonic anhydrase inhibitors 30.0% (*n* = 6/20), α_2_-agonists 5.0% (*n* = 1/20; Table 1).

The pre-operative questionnaire (T0) was completed and submitted by the patients 4.32 ± 2.01 months before the surgical procedure. All included patients were followed up by protocol and post-operative questionnaires were administered after 1 (T1) and 6 months (T2) from surgery. Pre- and post-operative outcomes are shown in Table 2 and Table 3.

Overall, at T0, 93.75% (*n* = 30/32) of eyes presented severe (Munk 3 or 4) epiphora and 68.75% (*n* = 22/32) had recurrent dacryocystitis. At T1, 50.0% of patients (*n* = 16/32) referred residual epiphora (Munk ≥ 1) and in 18.75% (*n* = 6/32) described symptoms of dacryocystitis. At T2, 31.25% (*n* = 10/32) of operated eyes were still affected by epiphora (Munk ≥ 1) and in just 6.25% of cases (*n* = 2/32) patients reported symptoms of lacrimal sac inflammation. With reference to right-sided procedures, at T0, 92.86% of patients (*n* = 13/14) were referred for relevant watering (Munk: 3–4), whereas 71.42% (*n* = 10/14) patients suffered from recurrent dacryocystitis. At T1, 42.85% (*n* = 6/14) patients reported residual epiphora (Munk ≥ 1), while only in 3 cases (21.42%) were inflammatory symptoms described; conversely, at T2, 35.71% of patients (*n* = 5/14) were still referred, at least for mild watering (Munk ≥ 1), and just 1 patient (3.12%) reported recurrent dacryocystitis. As regards left eye procedures, at T0, 94.44% of patients (*n* = 17/18) were suffering from severe lacrimation (Munk: 3–4) and 66.67% (*n* = 12/18) reported recurrent dacryocystitis. At T1, 44.44% of patients (*n* = 8/18) suffered from epiphora and 16.67% (*n* = 3/18) from symptoms of lacrimal gland inflammation. At T2, 27.78% of patients (*n* = 5/18) were still referred for epiphora, while just 1 (5.55%) was referred for recurrent dacryocystitis. The difference observed at aggregate timepoints, as well as between pre-operative (T0) and post-operative (T1 and T2) measurements, were statistically significant for paired and single-side analysis. On the other hand, there was no significance in the difference between T1 and T2 outcomes, with regard to both epiphora and dacryocystitis (Table 4).

Overall, at T1, 16 (50.0%) complete, 10 (31.25%) anatomical successes and 6 (18.75%) surgical failure were observed; likewise, at T2, there were 22 (68.75%) complete, 8 (25.0%) anatomical successes and 2 surgical failures (6.25%), which required revision surgery.

## 4. Discussion

The relationship between glaucoma and the lacrimal system has not yet been completely understood, but in recent years interest in the topic has risen. Both conditions are highly prevalent and can exert a substantial impact on the patient’s quality of life across clinical, social, and economic dimensions. According to certain estimates, glaucoma impacts over 2% of the general population aged 40 and above, with a prevalence of 10% among individuals aged 75 and older. It notably affects African-Caribbean individuals and demonstrates a higher occurrence in men compared to women [14,15]. On the other hand, Dalgleish conducted a study on a significant series of asymptomatic patients and discovered that around 9% of males and 10% of females aged 40 and above had nasolacrimal duct obstruction (NLDO) and/or lacrimal sac obstructions [16]. This finding revealed a much higher incidence than expected.

Chronic assumption of topical anti-glaucoma therapy has been shown to produce a chronic degree of subclinical inflammation (detected as increased expression of HLA-DR) on conjunctival epithelial cells [17]. This continuous inflammatory stimulus has been demonstrated to produce fibrotic changes on the conjunctival surface and on the upper lacrimal drainage system, as well as an overall impairment of Jones’ pump function [18,19,20]. Different clinical studies presented the effects of glaucoma eye drops on the eye and lacrimal system, inducing a widespread cicatricial reaction, which eventually turns into stenosis, and possibly in occlusion. Kashkouli et al. demonstrated that the puncti and canaliculi are the main anatomical sites of lacrimal drainage system obstruction associated with anti-glaucoma eye drops, and that both lower and upper obstruction were significantly more frequent in patients under topical anti-glaucoma medications [18]. On the other hand, Seider et al. reported that the irritative effect of chronically used timolol-containing topical glaucoma preparations can also potentially irritate the lacrimal mucosa located in the distal segment of the lacrimal system [19]. Their study revealed a significant increase in the incidence of NLDO among glaucoma patients compared to individuals without glaucoma-related conditions. This suggests that the irritative effects of timolol-containing eye drops may contribute to the development of NLDO in glaucoma patients. Moreover, Di Maria et al. showed that prostaglandin eye drops affect the outcome of EE-DCR in terms of increasing epiphora recurrence from early to late post-operative follow up [20]. McNab et al. investigated the time required for cicatricial remodeling of the conjunctiva and lacrimal ducts to occur following the initiation of eye drop administration. They presented a case series involving patients receiving topical ocular medications, with 43% prescribed anti-glaucoma drops, who developed lacrimal punctal and canalicular obstruction. The duration of exposure to these medications ranged from 3 weeks to 20 years. The authors’ findings suggested that lacrimal canalicular obstruction could manifest relatively soon after commencing topical ocular medications or could be part of a broader cicatricial reaction in patients on long-term medication regimens [9].

Our study cohort consisted of glaucoma patients diagnosed with NLDO. Despite acknowledging the possibility of dysfunction in the upper lacrimal duct among our patients, we chose to proceed with a posterior fossa EE-DCR approach as described by Metson [21]. Our goal was to maximize the improvement in the quality of life for these patients by addressing the impact of epiphora and recurrent dacryocystitis. Subsequently, we evaluated the efficacy of this approach. Our population was composed of 20 patients, for a total of 32 operated sides. This number meets the statistical minimum sample size and, while limited, it is nevertheless the largest series reported in the literature to date. Almost all patients in our sample were referred for significant pre-operative epiphora (Munk 3–4: 93.75%) and the majority (68.75%) also presented with at least occasional episodes of dacryocystitis. The surgical approach of choice for these cohort was the posterior EE-DCR, targeting the lacrimal sac through the thinnest lamina of the lacrimal bone and avoiding the need for a mucosal flap [22]. Both 1- and 6-month examinations showed significant surgical outcomes, with patients reporting a remarkable relief from pre-operative symptoms, confirming some efficacy of the procedure (Table 4). Additionally, even when patients reported persisting epiphora in T1 and T2 questionnaires, the Munk score severity of watering was gradually lower. In fact, although epiphora was referred in 50.0% of cases at T1 (*n* = 16/32) and in 31.25% (*n* = 10/32) at T2, Munk scores 3 and 4 were reported in 93.75% (*n* = 30/32) of eyes at T0, while only in 15.63% (*n* = 5/32) and 9.38% (*n* = 3/32) in T1 and T2, respectively (Table 2). This occurrence, in the presence of anatomical patency of the lower lacrimal system, may indeed be explained by some anti-glaucoma drops’ effects on the upper system and/or by the so called “functional block” [8].

According to previous reported series, EE-DCR in adults with primary acquired NLDO has an anatomical success varying from 89.7% to 98.1% [23,24,25], whereas the complete (or “functional”) success is reported as between 76.9% and 95.6% [26,27,28,29,30]. Our results depict a slightly worse scenario, our rate of anatomical and complete success being 93.75% and 68.75%, respectively, at T2 examination. It is opinion of the authors that the observed difference may be attributed to the damage caused by glaucoma to the nasolacrimal pump system and the upper lacrimal myo-contractile apparatus. Nevertheless, we stress that despite this damage, the effectiveness of DCR in these patients is not significantly diminished. Therefore, surgeons should not hesitate to recommend DCR in such cases.

Furthermore, another crucial aspect to consider are the outcomes achieved with EE-DCR in our patient cohort, concerning the incidence of recurrent dacryocystitis. Indeed, Rabina et al. highlighted in a retrospective study on ex-DCR how glaucoma patients had significantly higher odds (6.71-fold) of developing dacryocystitis compared to patients without glaucoma [31]. Similarly, within our series, nearly 70% of cases were reported to have recurrent dacryocystitis. However, it is noteworthy that surgical failure, characterized by persistent dacryocystitis, was observed in only two cases (6.25%) in our study, both of which necessitated revision surgery after EE-DCR. Even in the presence of dysfunction in the upper myo-contractile apparatus, it is likely that shortening the entire lacrimal drainage system could result in reduced pressure within the lacrimal sac. This decrease in pressure may ultimately alleviate fluid stagnation, reduce the synthesis of inflammatory mediators, and prevent bacterial overgrowth.

Hence, based on our findings, we are inclined to believe that a concurrent diagnosis of glaucoma should not deter a surgeon from conducting an EE-DCR procedure. This is because symptoms, such as epiphora and dacryocystitis, are expected to improve, aligning with success rates documented in the existing literature. Moreover, given that topical medications determine a slow and progressive damage effect on the upper lacrimal system, it would be interesting to evaluate in future studies if EE-DCR’s outcomes change with respect to time since first glaucoma diagnosis.

This study has several limitations. Firstly, its retrospective design may introduce bias into some of the outcome results. Moreover, despite being the largest case series reported in the literature on this topic and meeting the minimum sample size requirement, the study included only a limited number of patients. Thirdly, there was no comparison made with a control group comprising non-glaucoma patients with NLDO and it was not possible to retrieve the precise dosage and duration of anti-glaucoma medication used prior to surgery. Likewise, due to our limited sample size, we were not able to perform a comparison on post-operative outcomes between patients assuming different drug classes of anti-glaucoma eye drops. According to a recent analysis, glaucoma patients exposed to prostaglandin analogues showed a higher risk of developing post-surgical epiphora compared to patients exposed to other anti-glaucoma drug classes, with a greater difference at one year follow-up, possibly due to a higher proinflammatory effect in the long term [20]. Nevertheless, several evidence showed conflicting results. Cvenkel and Ihan demonstrated that β-blockers can induce a higher conjunctival inflammation than prostaglandin analogues, as evaluated by an increase in inflammatory marker expression and a decrease in mucin production [17]. Similarly, Guglieminetti et al. reported that patients receiving monotherapy with β-blockers would finally reach higher inflammatory levels compared to patients assuming topical prostaglandin analogues [32]. Furthermore, as concerns our sample, we didn’t notice relevant differences in post-operative outcomes based on the specific topical medication patients were into. Thereby, in order to ensure adequate statistical power and to meet the minimum sample size, we decided to include patients under different classes of topical anti-glaucoma medication. It is the opinion of the authors that, since this evidence needs to be strengthened, larger multi-centric prospective studies are warranted to accurately establish the effects of topical anti-glaucoma eye drops on patients undergoing EE-DCR for PANDO.

## 5. Conclusions

Based on our findings, it appears that EE-DCR in patients with NLDO who are also using topical anti-glaucoma medications is not contraindicated and may still be a viable therapeutic option. Indeed, at 6-month post-operative evaluation, only a limited subset of our entire sample (6.25%) still complained of a recurrence of dacryocystitis. In conclusion, despite the potential limitations posed by glaucoma eye drops on the functionality of the upper lacrimal drainage system, we may speculate that EE-DCR could provide substantial clinical relief from epiphora and significantly reduce the incidence of recurrent dacryocystitis. This improvement in symptoms might contribute to an enhanced quality of life for patients affected by these conditions, which will eventually be an area for further research.

## Figures and Tables

**Figure 1 jpm-14-00348-f001:**
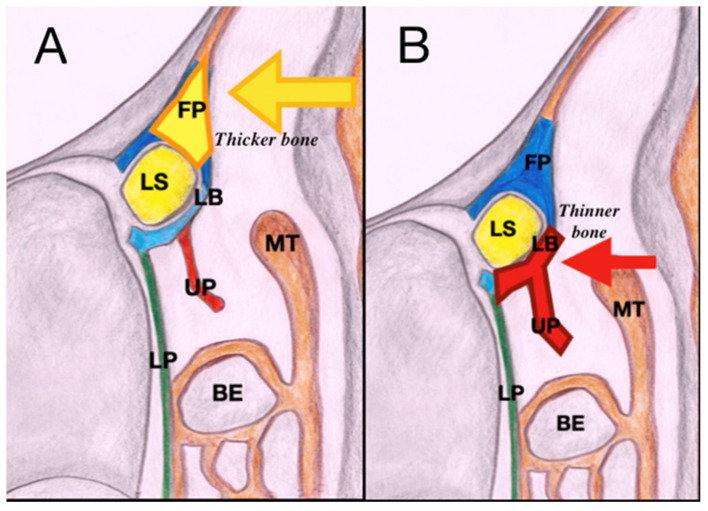
(**A**) Anterior EE-DCR is performed through a wide osteotomy and drilling of the frontal process of the maxilla. (**B**) Posterior approach, after uncinectomy, encompasses the drilling of a thinner area at the level of the lacrimal bone. In this way, preparing mucosal flaps is not necessary since the sac is approached at its thinnest walls, without leaving any area of bone exposed. (Abbreviations: LS, lacrimal sac; LB, lacrimal bone; FP, frontal process of the maxilla; MT, middle turbinate; LP, lamina papyracea, UP, uncinate process; BE, bulla ethmoidalis).

**Table 1 jpm-14-00348-t001:** Anti-glaucoma eye drops assumed by enrolled patients, by active principle.

Patient	Anti-Glaucoma Eye Drop	
	Left Eye	Right Eye
#1	Latanoprost	
#2	Latanoprost, timolol	Latanoprost, timolol
#3	Brinzolamide, brimonidine	
#4	Travoprost	
#5	Latanoprost, timolol	Latanoprost, timolol
#6	Brinzolamide, timolol	
#7	Brinzolamide, timolol, bimatoprost	Brinzolamide, timolol, bimatoprost
#8	Latanoprost, timolol	Latanoprost, timolol
#9	Dorzolamide, timolol	Dorzolamide, timolol
#10	Travoprost, timolol	Travoprost, timolol
#11	Brinzolamide, timolol	Brinzolamide, timolol
#12	Brinzolamide, timolol	Brinzolamide, timolol
#13		Tafluprost, brinzolamide, timolol
#14	Tafluprost, timolol	Tafluprost, timolol
#15	Dorzolamide, timolol	Dorzolamide, timolol
#16	Travoprost, timololo	Travoprost, timololo
#17	Dorzolamide, timolol	Dorzolamide, timolol
#18	Latanoprost, timolol	
#19		Latanoprost
#20	Dorzolamide, timolol	

**Table 2 jpm-14-00348-t002:** Epiphora at different timepoints by operated side.

	Munk Score	Overall (%)	Left (%)	Right (%)
T0	0	2 (6.25)	1 (5.55)	1 (7.14)
	1	0 (0)	0 (0)	0 (0)
	2	0 (0)	0 (0)	0 (0)
	3	6 (18.75)	4 (22.22)	2 (14.28)
	4	24 (75.0)	13 (72.22)	11 (78.57)
T1	0	16 (50.0)	8 (44.44)	8 (57.14)
	1	8 (25.0)	3 (16.66)	3 (21.43)
	2	3 (9.37)	2 (11.11)	1 (7.14)
	3	2 (6.25)	1 (5.55)	1 (7.14)
	4	3 (9.37)	2 (11.11)	1 (7.14)
T2	0	22 (68.75)	11 (61.11)	8 (57.14)
	1	4 (12.5)	3 (16.67)	1 (7.14)
	2	3 (9.38)	1 (5.56)	2 (14.29)
	3	1 (3.12)	0 (0)	1 (7.14)
	4	2 (6.25)	1 (5.56)	1 (7.14)

**Table 3 jpm-14-00348-t003:** Dacryocystitis at different timepoints by operated side.

		Overall (%)	Left (%)	Right (%)
T0	No	10 (31.25)	6 (33.33)	4 (28.57)
	Occasionally (<one/month)	10 (31.25)	5 (27.78)	5 (35.71)
	Often (>one/month)	12 (37.5)	7 (38.89)	5 (35.71)
T1	No	26 (81.25)	15 (83.33)	11 (78.57)
	Occasionally (<one/month)	3 (9.37)	2 (11.11)	1 (7.14)
	Often (>one/month)	3 (9.37)	1 (5.55)	2 (14.28)
T2	No	30 (93.75)	17 (94.44)	13 (92.86)
	Occasionally (<one/month)	0 (0)	0 (0)	0 (0)
	Often (>one/month)	2 (6.25)	1 (5.55)	1 (7.14)

**Table 4 jpm-14-00348-t004:** Difference in clinical outcomes at aggregate and paired timepoints. Results are presented in form of *p*-value.

Timepoint	Epiphora			Dacryocystitis			
	Overall	Left	Right	Overall	Left	Right	
Aggregate	<0.001	<0.001	<0.001	<0.001	0.001	<0.001	<0.001
T0 vs. T1	<0.001	0.001	0.006	0.001	0.034	0.008	<0.001
T0 vs. T2	<0.001	<0.001	0.002	<0.001	0.017	0.003	<0.001
T1 vs. T2	0.431	0.596	0.695	0.694	0.791	0.769	0.431

## Data Availability

Data are contained within the article and Appendix A.

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
