# Peer review of "Outcomes of Endoscopic Endonasal Dacryocystorhinostomy in Glaucoma Patients"

_jpm, 2024, doi:10.3390/jpm14040348_

Round 1

Reviewer 1 Report

Comments and Suggestions for Authors

It is a interesting study.

I evaluated the article entitled Outcomes of endoscopic endonasal dacryocystorhinostomy in 2

glaucoma patients and I have the following comments:

Introduction. Well-documented, but it would be interesting to mention more studies related to glaucoma vs DCRS relationship

Material and method.

Lines 99. Perhaps to inclusion criteria if it can be specified how many antiglaucoma drugs he puts and for how long.

Which was coded vis-à-vis antiglaucoma medication after surgery.

Results: no reference to glaucoma or antiglaucoma medication. The recurrence of dacryocystitis is the sea detul.

Conclusions. It can be improved. It should reflect the results

Kind regards

Reviewer 2 Report

Comments and Suggestions for Authors

While being a retrospective study with all the inherent risks for biases, I think it addresses an interesting point, which warrants its publishing.

As the authors state, it appears to be the largest case series of EE-DCR in patients that have previously been using topical antiglaucoma medication. The study proves that EE-DCR remains an effective surgery in this particular type of patients. 

I only have a few comments:

line 44 - "Secondary nasolacrimal duct obstruction (SANDO) is defined whenever a specific inflammatory event is responsible" - one can argue that neoplasia or trauma may not include necessarily an inflammatory event

It would be interesting to know how long was the antiglaucoma medication used before the surgery (however, being a retrospective study, this information may not be available).

line 153  - please explain the meaning of the term "eye dubbing"

Tables 2 and 3 - I do not support reporting the success by the operated side (left or right). It does not bring additional useful information.

lines 196-197 - "lacrimal gland inflammation' - do you mean "lacrimal sac inflammation' ?
